# Education for Sustainable Development in Spanish University Education Degrees

**Fermín Sánchez-Carracedo** [1] , **Francisco Manuel Moreno-Pino** [2,*] , **Daniel Romero-Portillo** [3] and **Bárbara Sureda** [1]

1   University Research Institute for Sustainability Science and Technology, Universitat Politècnica de Catalunya-BarcelonaTech, 08034 Barcelona, Spain; fermin.sanchez@upc.edu (F.S.-C.); barbara.sureda@upc.edu (B.S.)
2   Didactics Department, Universidad de Cádiz, 11519 Cádiz, Spain
3   Sociology Department, Universidad Pablo de Olavide, 41013 Sevilla, Spain; drompor@upo.es
*   Correspondence: franciscomanuel.moreno@uca.es

**Abstract:** This work presents an analysis of student perception of Spanish university education degrees regarding their training in sustainable development. A sample of 942 students was used. The methodology consists of analyzing the results of a survey answered by the first- and fourth-year students from nine education degree courses in four Spanish universities. Comparison of the perception of learning by fourth-year students against those of the first year enables improvements in learning regarding sustainability to be ascertained. The questionnaire consists of 18 questions concerning four sustainability competencies: C1-Critical contextualization of knowledge, C2-Sustainable use of resources, C3-Participation in community processes, and C4-Ethics. Two composite indicators are defined to analyze the absolute learning (achieved on completion of their studies) and the relative learning (achieved with respect to what should have been achieved) declared by the students in each competency, degree and university. The results show that students declare an improvement in all their sustainability competencies, although the results of the final learning are far from those expected: they have learned only 27% of what they should have learned. Moreover, the learning achieved in the four competencies depends on the degree and the university.

**Keywords:** sustainability; sustainable development; education for sustainable development; Spanish higher education; students' learning; students' perception; sustainability competencies; EDIN-SOST project





## 1. Introduction

The current educational system was conceived in the eighteenth century in the intellectual culture of the Enlightenment, and developed in the economic circumstances of the industrial revolution during the nineteenth century. It was consolidated during the twentieth century in France, England and Germany, whose educational systems were totally or partially imitated by other countries [1]. The material and economic conditions of nineteenth-century society, strongly driven by industrialization processes, reinforced the enlightened idea of public education, thereby establishing the principle of universal, free and compulsory education in elementary schools [2].

With the arrival of industrialization, oil, natural gas and coal became the three basic non-renewable fuels of the current energy model. The serious pollution derived from their production, together with the quasi-exponential growth of population–consumption factors in recent years, have led to important problems (poverty, inequality, climate change, etc.) [3].

The holistic systemic effect and the complex nature of these problems require a paradigm shift if our determination to solve them is to be effective. The speed and acceleration of the complex processes at work in our planetary era add their own uncertainties, and urgent action is necessary [4].

Sustainable development is presented as an alternative to classic development models. The 2030 Agenda has an integrating vision of sustainable development. In fact, the Sustainable Development Goals (SDGs) are a widely accepted framework for promoting sustainable development [5]. SDG4 goal 4.7 pursues the "sustainability" of education, understood as the design of training programs in the different areas of knowledge, with the aim of providing citizens with the theoretical and practical knowledge necessary to promote sustainable development [6].

Changes in the behavior of people, institutions and organizations are a prerequisite for sustainable development [7]. In this sense, the role of higher education is key to promoting the transition towards a more sustainable society [8].

The inclusion of sustainability principles in university curricula should not be reduced to the incorporation of sustainability content in the syllabus of a subject, but implies for teachers a change in attitude, methodology and conception of teaching-learning processes that address the socio-environmental and socio-constructivist dimension. Teacher-training based on participatory learning and ethical considerations may contribute to stimulation of motivation when addressing the sustainability challenge in the classroom [9]. The entire educational process must be approached from a holistic perspective [10], and sustainable education practice extended to be more systemic and cross-disciplinary [11]. Future graduates need to be provided with competencies-based training [12]. The competencies approach is associated with holistic, complex and above all dialogical development processes. Such an approach is shown to provide an opportunity for knowledge creation and the transfer of sustainability issues in a democratic and emancipatory way [13].

Some authors have investigated which competencies should be included in university curricula for the introduction of sustainable development, whether or not these competencies should be transversal, and what their degree of integration is [14]. Wiek et al. [15] identify the following competencies as being related to sustainability: systems-thinking, anticipatory, normative, and strategic and interpersonal competency. UNESCO [16] adds critical thinking, self-awareness and integrated problem-solving.

Other authors have analyzed the inclusion of environmental competencies in the bachelor's degree in Primary Education taught in 23 Spanish universities, highlighting the lack of specific training in environmental education in the curricula [17].

In the Spanish context, the CRUE (Conference of Rectors of Spanish Universities) sets out general criteria, recommends actions for the introduction of sustainability into university curricula, and proposes four transversal sustainability competencies that must be integrated into university training [10]. The EDINSOST project [18] analyzed the CRUE sustainability competencies that teachers and students in the Spanish university system possess.

## 2. Materials and Methods

### 2.1. Objective, Research Questions and Starting Hyphoteses

The objective of this work is to present a methodology to measure the perception of learning in sustainability received by students of a given degree. This paper presents the results of the EDINSOST project regarding the sustainability competencies held by the students of nine Spanish university degrees in Education, and how the students perceive the evolution of these competencies throughout their studies. The nine degrees are taught at four Spanish universities.

This paper seeks to answer the following research questions:

- Q1: How much do students from Spanish university education degrees in Spain perceive the improvement in their sustainability competencies during their studies?
- Q2: Is the improvement homogeneous at all domain levels?
- Q3: Is the improvement homogeneous in all the degrees analyzed?
- Q4: Is the improvement homogeneous in all the universities analyzed?
- Q5: Does the same degree provide homogeneous learning in all the universities where it is taught?

- Q6: Does the same university provide homogeneous learning in all the degrees it teaches?

These questions are operationalized in the following starting hypothesis: fourth-year students have improved their sustainability competencies compared to first-year students. In this paper, competencies improvement will be indirectly measured on the basis of the students' perceived learning along their training process. However, the same methodology can be used with a questionnaire that assesses actual student learning, rather than perceived learning.

### 2.2. Instruments

The EDINSOST sustainability questionnaire for education degrees [19] has been used to answer these research questions. The questionnaire can be consulted in Appendix A at the end of the paper. It has undergone a rigorous validation process [20], and is based on the sustainability map of education degrees [21].

A sustainability map is a competencies map [22] that develops sustainability competencies. The sustainability map of education degrees contains the learning outcomes related to sustainability that students of an education degree must have acquired on completion of their studies. Learning outcomes are classified according to a learning taxonomy. In the case of the EDINSOST project, a simplified version of Miller's pyramid is used as the taxonomy [23,24]. Miller's pyramid is a four-level taxonomy ("Know", "Know-how", "Demonstrate and Do") defined for the field of medicine. In this field, it is important to differentiate the "Demonstrate and Do" levels, since doctors must train ("Demonstrate") before operating ("Do"), as human lives are at stake. By transferring this taxonomy to the field of education, and with the aim of reducing the number of levels of the taxonomy, we decided to unify the "Demonstrate" and "Do" levels into a single level.

The learning outcomes are defined on the basis of the sustainability competencies as stated by the CRUE [10]. These competencies must be developed in all Spanish higher studies, regardless of their field. Each CRUE Competency is defined more precisely and from a holistic perspective in terms of Competency Units (CUs). Table 1 shows the four sustainability competencies defined by the CRUE and the CUs of the sustainability map of the education degrees.

The sustainability map [19] is made up of 18 cells (six competency units classified into three domain levels). Each cell contains a unique learning outcome. The EDINSOST questionnaire contains a proposal for each learning outcome (see [19]). Proposals are answered using a 4-point Likert scale: "Strongly disagree", "Disagree", "Agree" and "Strongly agree". An even number of points on the Likert scale have been chosen to induce students to position themselves towards the option "agree" or "disagree", thus avoiding the existence of a neutral response. The number of points was selected from a validation process carried out by experts [21], who judged that the students would not be able to discriminate correctly between four and six points, and four points would therefore be sufficient.

### 2.3. Sample

The questionnaire was addressed to four degree courses belonging to four universities:

- Universities

UCA: University of Cádiz
UIC: International University of Catalonia
US: University of Seville
USAL: University of Salamanca

- Degrees

BDSE: Bachelor's Degree in Social Education
BDECE: Bachelor's Degree in Early Childhood Education
BDPE: Bachelor's Degree in Primary Education
BDP: Bachelor's Degree in Pedagogy

**Table 1.** Conference of Rectors of Spanish Universities (CRUE) Sustainability competencies and Competency Units (CUs) of the sustainability map of education degrees in the EDINSOST project, as presented in Muñoz-Rodríguez et al. [25].

| Competency | Competency Units |
|---|---|
| C1. Critical contextualization of knowledge by establishing interrelations with social, economic, environmental, local, and/or global problems. | CU1.1 Understands the functioning of natural, social, and economic systems, as well as their interrelations and problems, both at a local and global level. |
| | CU 1.2 Possesses critical thinking and creativity, taking advantage of the different opportunities presented (ICT, strategic plans, regulations, etc.) in the planning of a sustainable future. |
| C2. Sustainable use of resources and prevention of negative impacts on the natural and social environment. | CU 2.1. Designs and develops actions, making decisions that take into account the environmental, economic, social, cultural, and educational impacts so as to improve sustainability. |
| C3. Participation in community processes that promote sustainability. | CU 3.1 Promotes and participates in community activities that encourage sustainability. |
| C4. Application of ethical principles related to the values of sustainability in personal and professional behavior. | CU 4.1. Is consistent in actions respecting and valuing (biological, social, and cultural) diversity and committed to improving sustainability. CU 4.2. Promotes education in values oriented to the formation of responsible, active and democratic citizens. |

The questionnaire was answered using Google Forms during the second semester of the 2018 academic year, and the target consisted of the first- and fourth-year students in each degree. The total sample size is 942, 548 first-year students and 394 fourth-year students. Some degrees are not taught in some universities, so they have not been taken into account in this study. Degrees in which a significant number of responses were not obtained have also not been considered. Table 2 shows a break-down of the sample used in this study and response rate (percentage of students who answered the questionnaire over the total number of students enrolled) according to university, course and degree.

**Table 2.** Sample and response rate broken down by University, course and degree.

| | | Academic Degrees | | | | |
|---|---|---|---|---|---|---|
| Universities | Course | Bachelor's Degree in Social Education (BDSE) | Bachelor's Degree in Early Childhood Education (BDECE) | Bachelor's Degree in Primary Education (BDPE) | Bachelor's Degree in Pedagogy (BDP) | Overall |
| University of Cádiz (UCA) | 1st | 0 | 37 (20.7%) | 142 (52.6%) | 0 | 179 (36.6%) |
| | 4th | 0 | 61 (70.3%) | 123 (86.9%) | 0 | 184 (78.6%) |
| International University of Catalonia (UIC) | 1st | 0 | 17 (51.5%) | 18 (40.9%) | 0 | 35 (46.2%) |
| | 4th | 0 | 14 (37.8%) | 14 (27.5%) | 0 | 28 (32.6%) |
| University of Seville (US) | 1st | 0 | 0 | 104 (20.3%) | 0 | 104 (20.3%) |
| | 4th | 0 | 0 | 86 (9.1%) | 0 | 86 (9.1%) |
| University of Salamanca (USAL) | 1st | 53 | 71 (86.6%) | 51 (81.0%) | 55 (78.6%) | 230 (75.5%) |
| | 4th | 51 | 9 (10.1%) | 26 (22.0%) | 10 (14.3%) | 96 (28.6%) |
| Overall | | 104 | 209 (46.2%) | 564 (42.5%) | 65 (46.4%) | 942 (100%) |

*2.4. Methodology*

Two composite indicators have been created that facilitate the interpretation without undermining its validity, reliability or precision [26]. Using composite indicators as a statistical manipulation technique reduces the complexity of the 18 questions in the questionnaire [27]. In this paper, composite indicators are built for each analysis dimension: competencies, competency units and domain levels.

The first composite indicator, Learning Increase, measures the absolute learning that students perceive from first to fourth course. The construction of this indicator consists of three stages.

- First, the structure of the data matrix was explored. Due to the presence of asymmetries, imputations of the mean were made to the missing values and the distributions were standardized [26].
- Second, the one-dimensionality of the data structure and the reliability of the measurements were studied throughout the whole questionnaire. The objective of this process is to measure, on the one hand, to what extent all the variables that are added in a composite indicator are actually measuring the same underlying construct and, on the other hand, to what extent these measurements are internally consistent. One-dimensionality analysis was performed through KMO and Bartlett sphericity tests. KMO value for the whole questionnaire is 0.939, and Bartlett sphericity test is significant ($\chi^2 = 9834.271$; $p = 0.000$), thus indicating data structure is suitable for performing a factor analysis. Reliability shows Cronbach's alpha coefficient is 0.9368, verifying that the set of questions of the questionnaire is internally consistent. All these tests indicate that it is both possible and feasible to reduce the original questions or variables to composite indicators. Next, a Principal Components Analysis (PCA) was performed for each composite indicator representing competencies, competency units and domain levels. The Kaiser Criterion [28], which only considers main components whose eigenvalues are greater than 1, was used in PCAs. In all cases, the first of the main components satisfied this criterion and, in addition, were able to account for more than 60% of the variance of the original variables. Finally, the factorial scores of the selected components were obtained in the PCAs.
- Third, in order to facilitate the interpretation and use of the composite indicators, a rescaling was performed to adjust the indicators to the unit of measurement of the responses, 0–3 [26,29]. We call this indicator Learning Increase, and it is calculated using Equation (1):

$$\frac{CI - min_{CI}}{max_{CI} - min_{CI}} * 3, \tag{1}$$

CI being the composite indicator.

The second indicator, Learning Percentage, is a measure of relative learning that was built from the Learning Increase indicator. This indicator measures the increase in aggregate learning of fourth-year students compared to first-year students, based on the learning they still had to acquire. This measure is aggregated, since the sample consists of a cross-sectional study (and not a study panel). This fact precludes the measurement of change at the subject level, but not for each degree, university, or for all students. The rescaling formula used is shown in Equation (2):

$$\frac{AL_4 - AL_1}{3 - AL_1}, \tag{2}$$

$AL_4$ being the 4th course university/degree-aggregated learning, and $AL_1$ the 1st course university/degree-aggregated learning.

As shown in Table 2, the percentage of responses is very uneven with respect to the number of students enrolled according to degree and university. To analyze whether the response rate has any effect on the results, a correlational analysis was carried out between

the two composite indicators and the response frequencies of each course, university and degree. Table 3 presents these data. In general, it is observed that the lower the number of responses, the greater the knowledge declared in each of the competencies and competency units. Furthermore, it is observed that this relationship is more intense and more significant for the Learning Increase indicator.

**Table 3.** Correlational analysis between declared knowledge and responses to the questionnaire.

| Competencies/Competency Units | Survey Response Rate | Gross Number of Participants |
|:---:|:---:|:---:|
| C1 | −0.3643 | −0.5031 * |
| CU1.1 | −0.3244 | −0.2705 |
| CU 1.2 | −0.4047 + | −0.5757 * |
| C2 | −0.4459 + | −0.5162 * |
| C3 | −0.4466 + | −0.4405 + |
| C4 | −0.5078 * | −0.6154 ** |
| CU 4.1 | −0.4056 + | −0.6353 ** |
| CU 4.1 | −0.5185 * | −0.5361 * |

$+ \, p < 0.1$; $* \, p < 0.05$; $** \, p < 0.01$.

Values represent Pearson r except for C1 and UC4.1, which are not normally distributed. These values represent Spearman rho.

## 3. Results

This section provides six figures which provide answers to the six research questions. Figures are discussed in Section 4. Figures consist of two grouped bar graphs: "a" and "b". On a scale of 0 to 3 (Likert scale of the questionnaire), the figures labeled as (a) show the value of the first composite indicator, Learning Increase, for first- and fourth-year students. The Y axis is called "Learning Increase" because the graph shows the learning differences between the first- and fourth-year students. On a normalized scale from 0 to 1, the figures labeled as (b) show the value of the second composite indicator, "Learning Percentage". This indicator shows the percentage of learning achieved by students in the fourth year compared to those in the first year (the value 1 represents the maximum learning they could have achieved).

Figure 1 shows the learning declared by all students in each competency and on average.

Figure 2 shows the learning reported by students broken down by CUs. Since competencies C2 and C3 only have one CU, the data shown for these two competencies is identical to that shown in Figure 1.

Figure 3 shows the learning declared by the students in each competency at each domain level of the learning taxonomy: L1-Know, L2-Know how and L3-Demonstrate and Do.

Figures 4 and 5 enable us to answer the third research question. Both figures present learning according to degrees, by competencies (Figure 4) and competency units (Figure 5).

To answer the research question Q4, Figure 6 shows the learning declared by students in each competency at each university.

Figure 7 enables us to answer the fifth research question. Data are only presented from BDECE and BDPE, because they are the two degrees that are taught at more than one university. BDSE and BDP are taught only at USAL and have been previously analyzed in Figures 4 and 5.

Figure 8 shows the learning declared by students in the four competencies in each university according to the degree, and allows us to answer research question Q6.

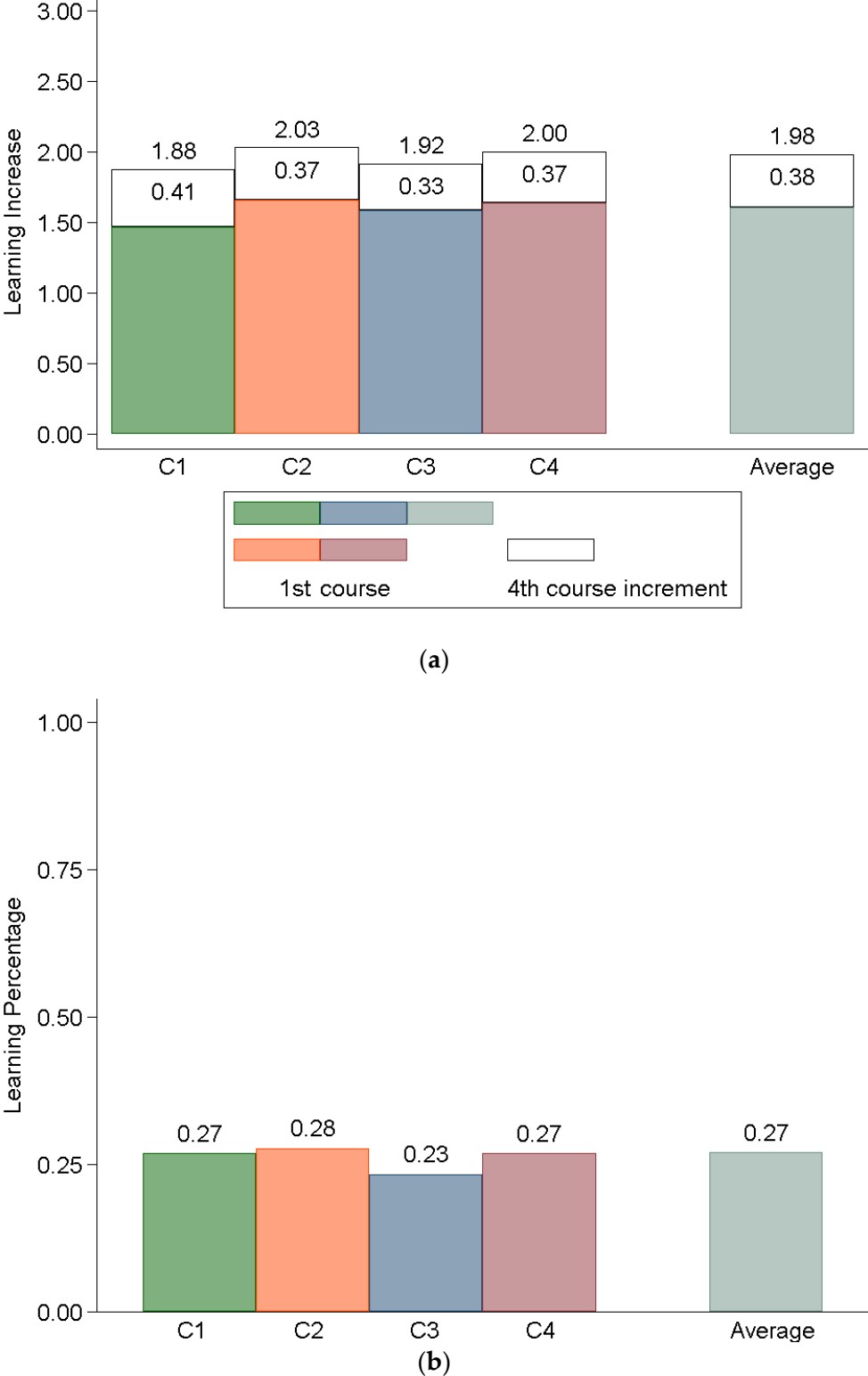

**Figure 1.** Learning Increase (**a**) and Learning Percentage (**b**) declared by the students in each competency and on average.

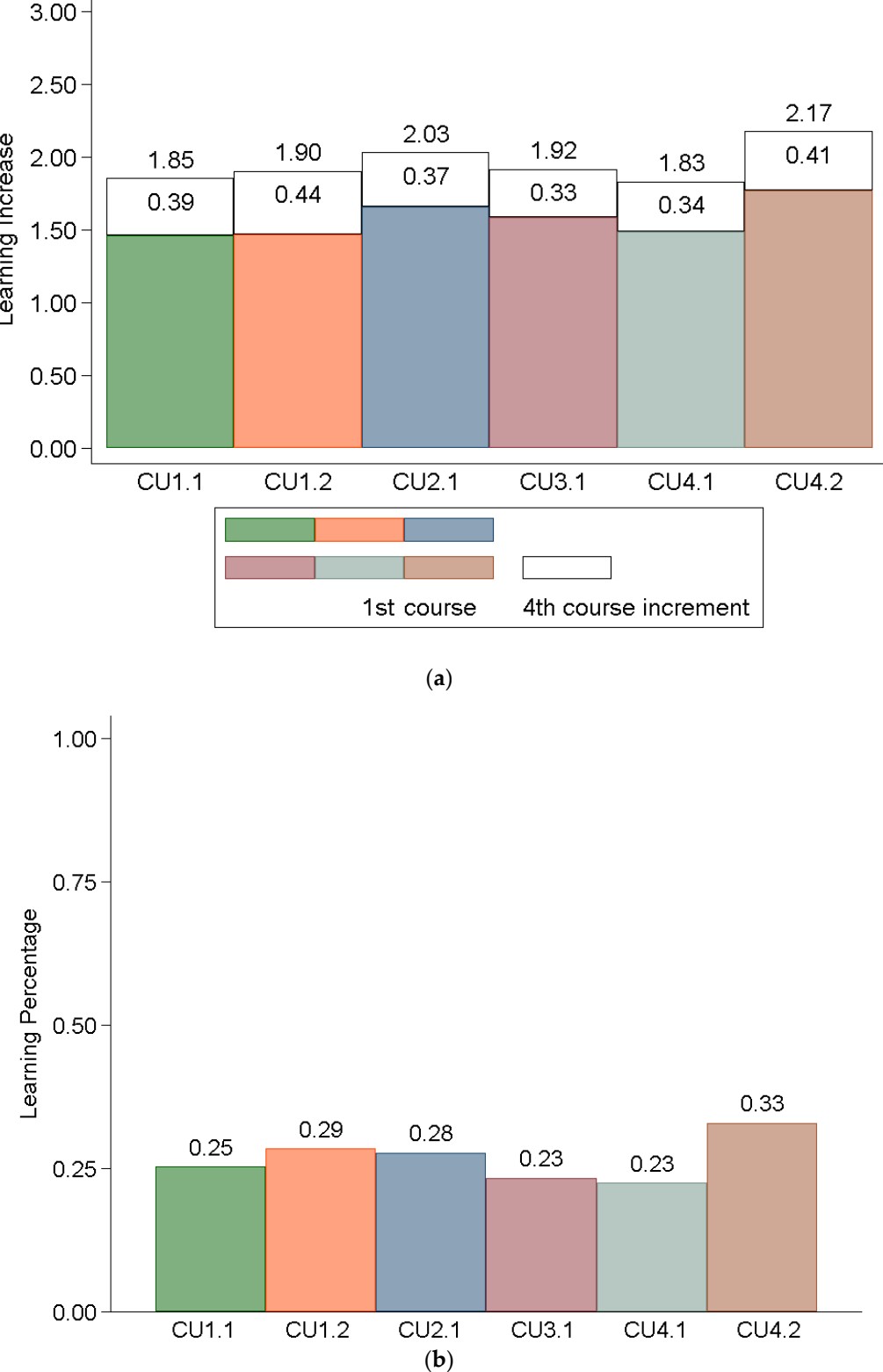

**Figure 2.** Learning Increase (**a**) and Learning Percentage (**b**) declared by the students in each Competency Unit.

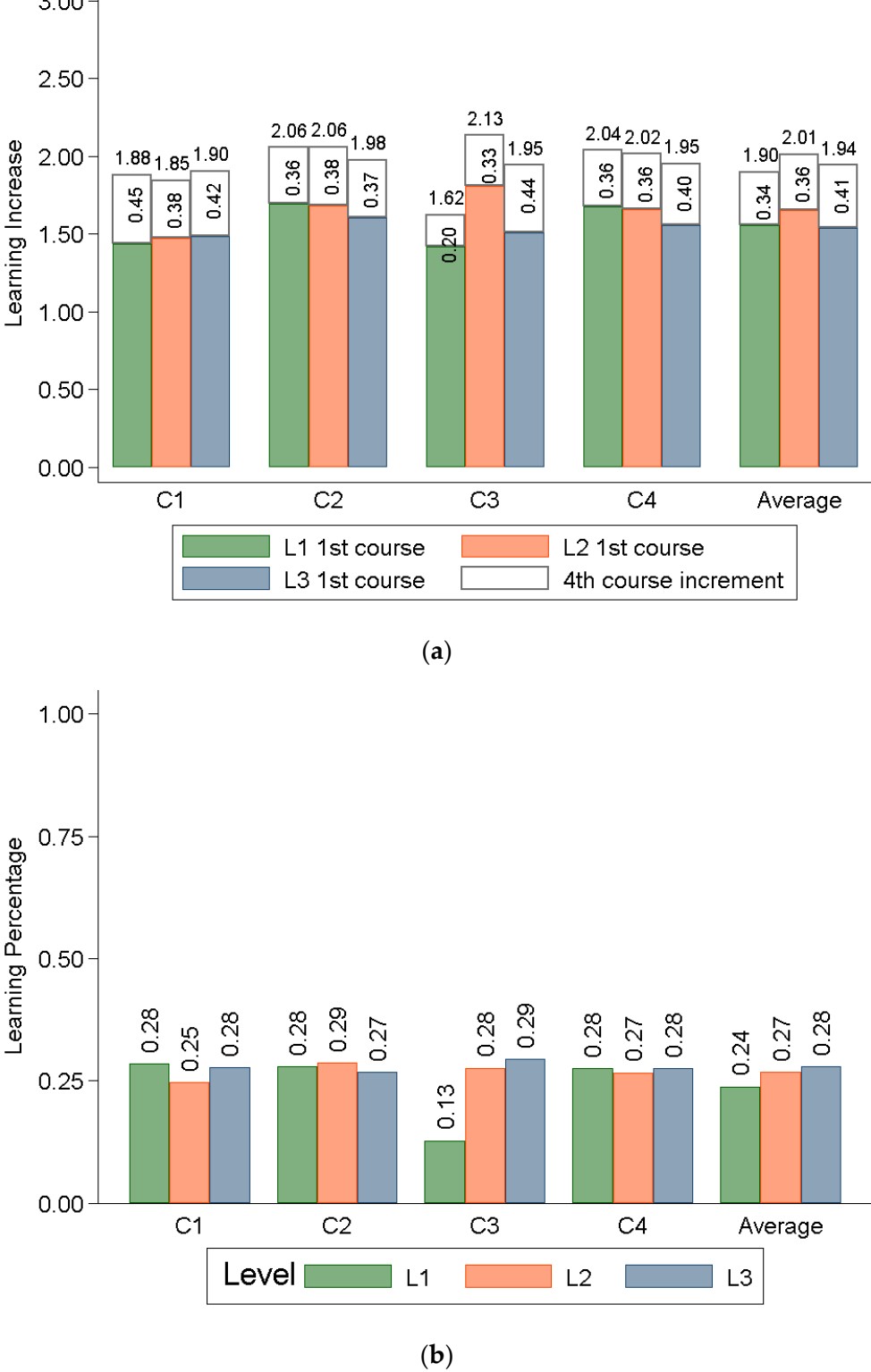

**Figure 3.** Learning Increase (**a**) and Learning Percentage (**b**) declared by the students at each domain level for each competency.

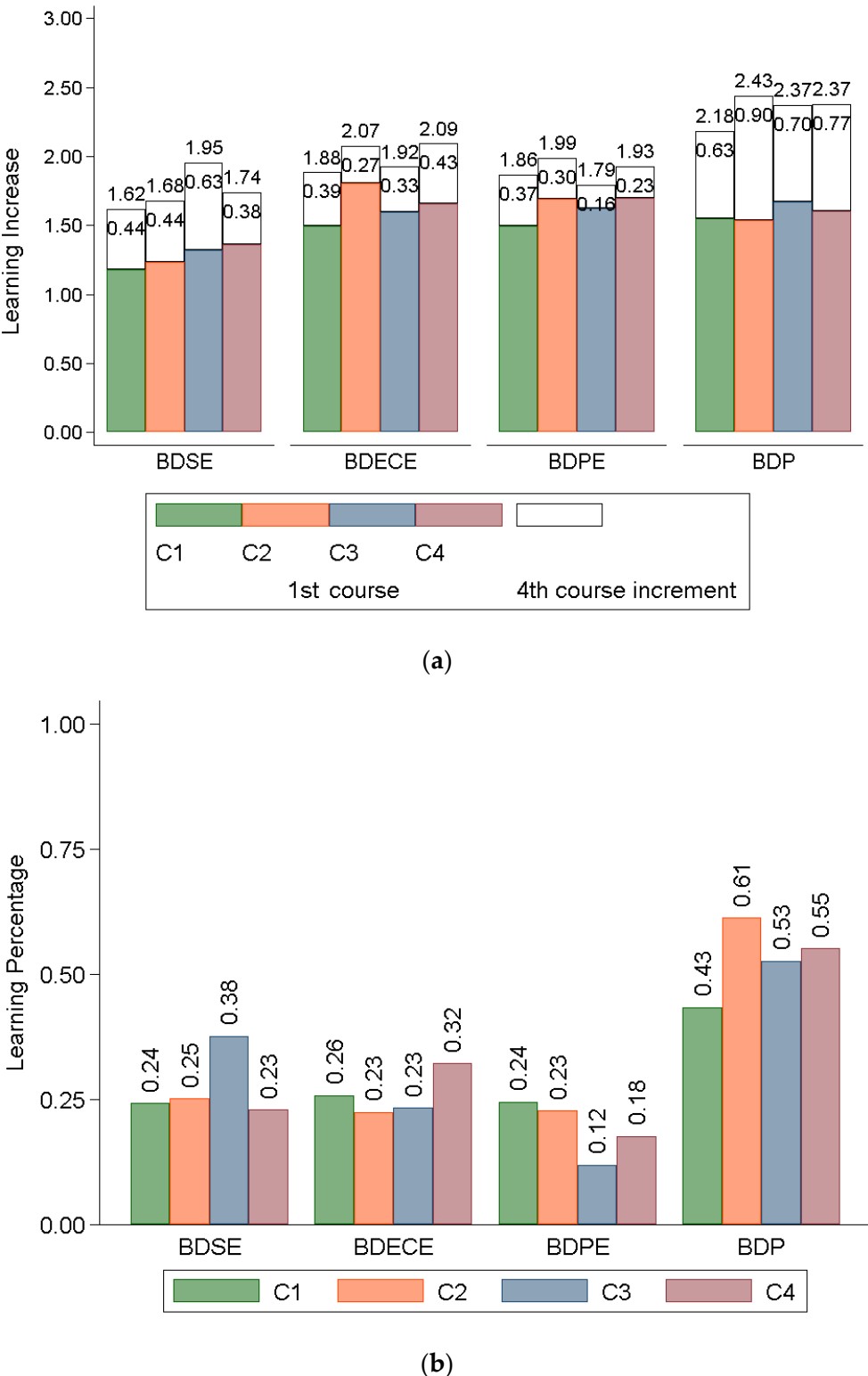

(**a**)

(**b**)

**Figure 4.** Learning Increase (**a**) and Learning Percentage (**b**) declared by the students in each competency for each degree.

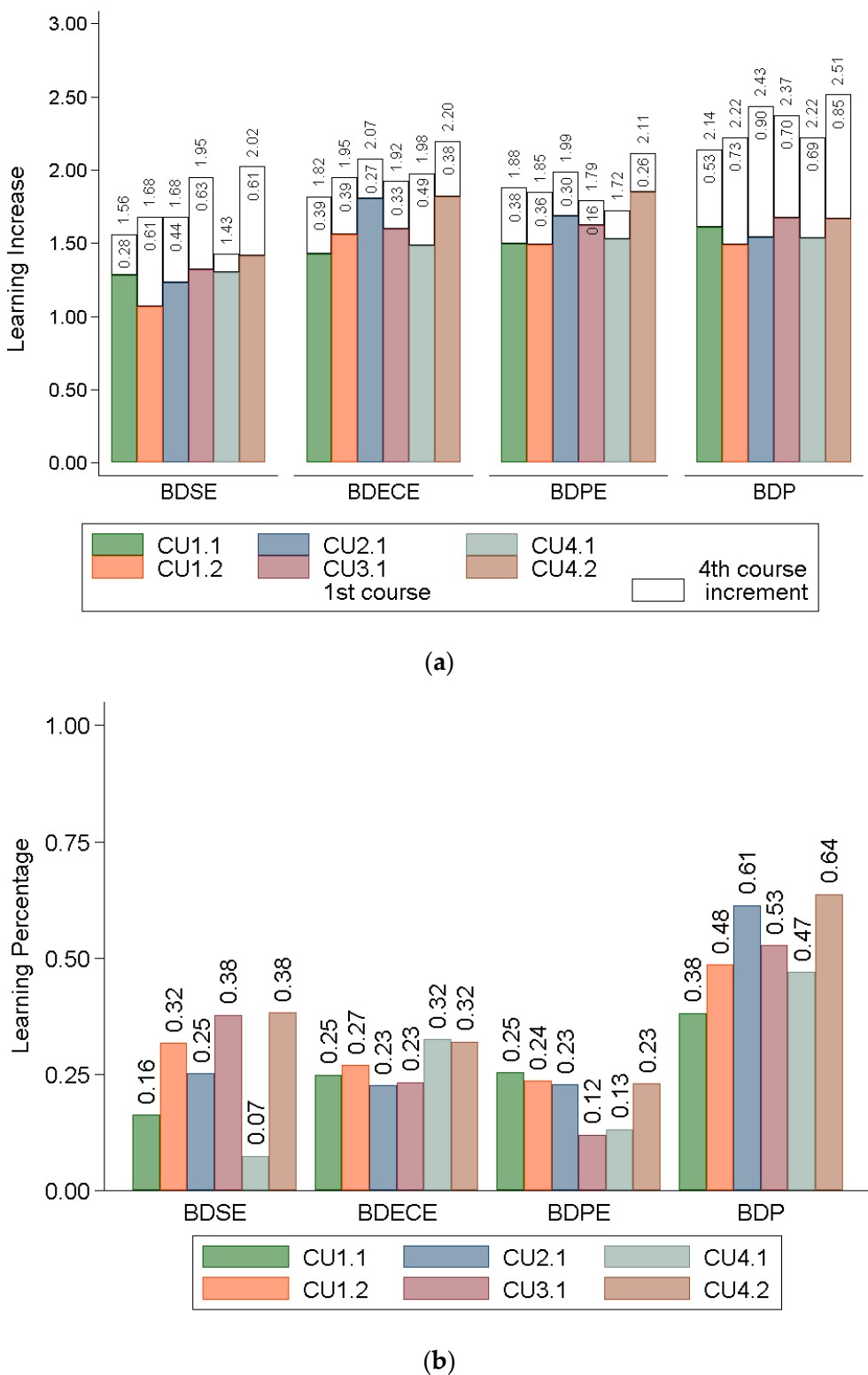

(**a**)

(**b**)

**Figure 5.** Learning Increase (**a**) and Learning Percentage (**b**) declared by the students in each competency unit in each degree.

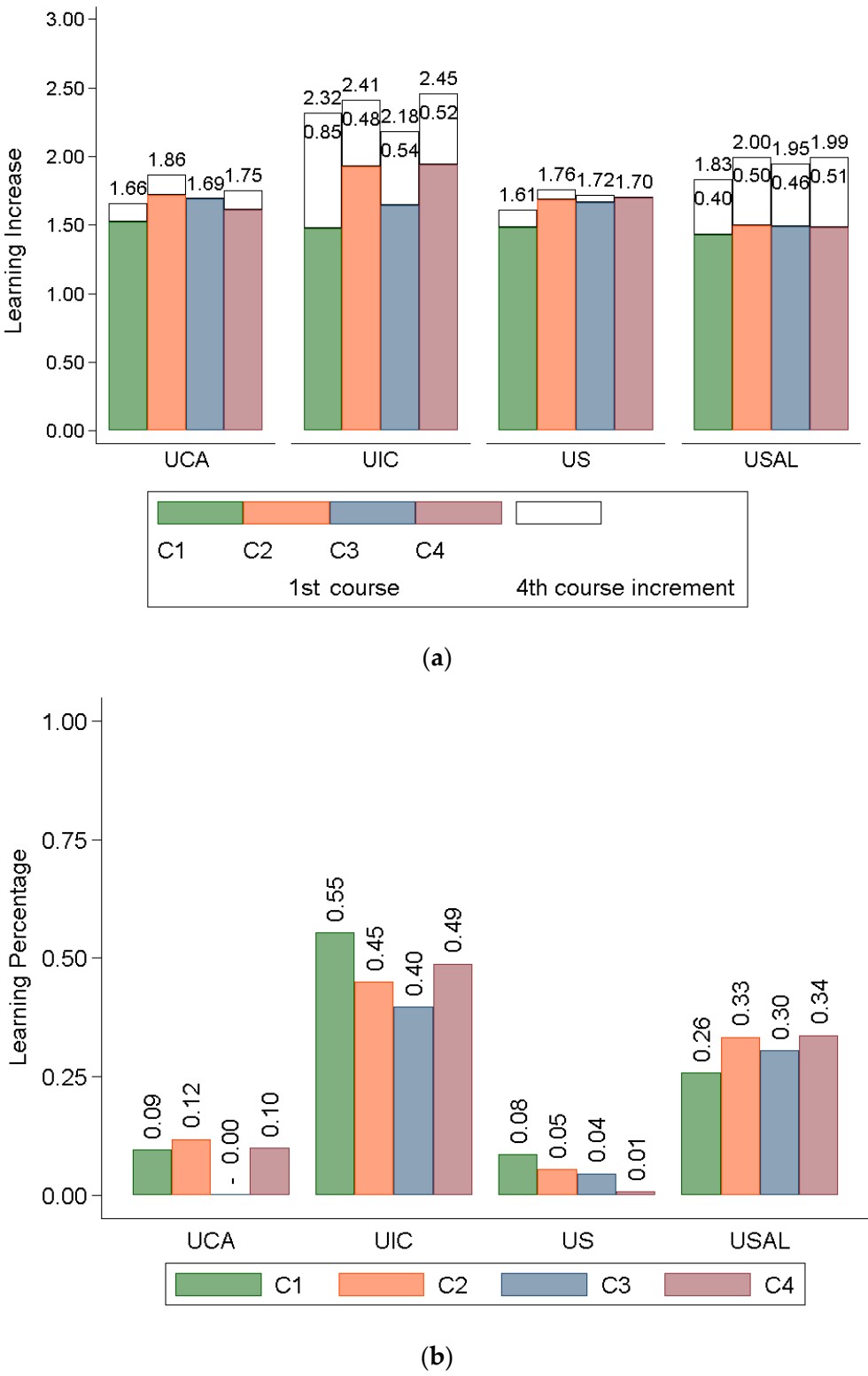

(**a**)

(**b**)

**Figure 6.** Learning Increase (**a**) and Learning Percentage (**b**) declared by the students in each competency and in each university.

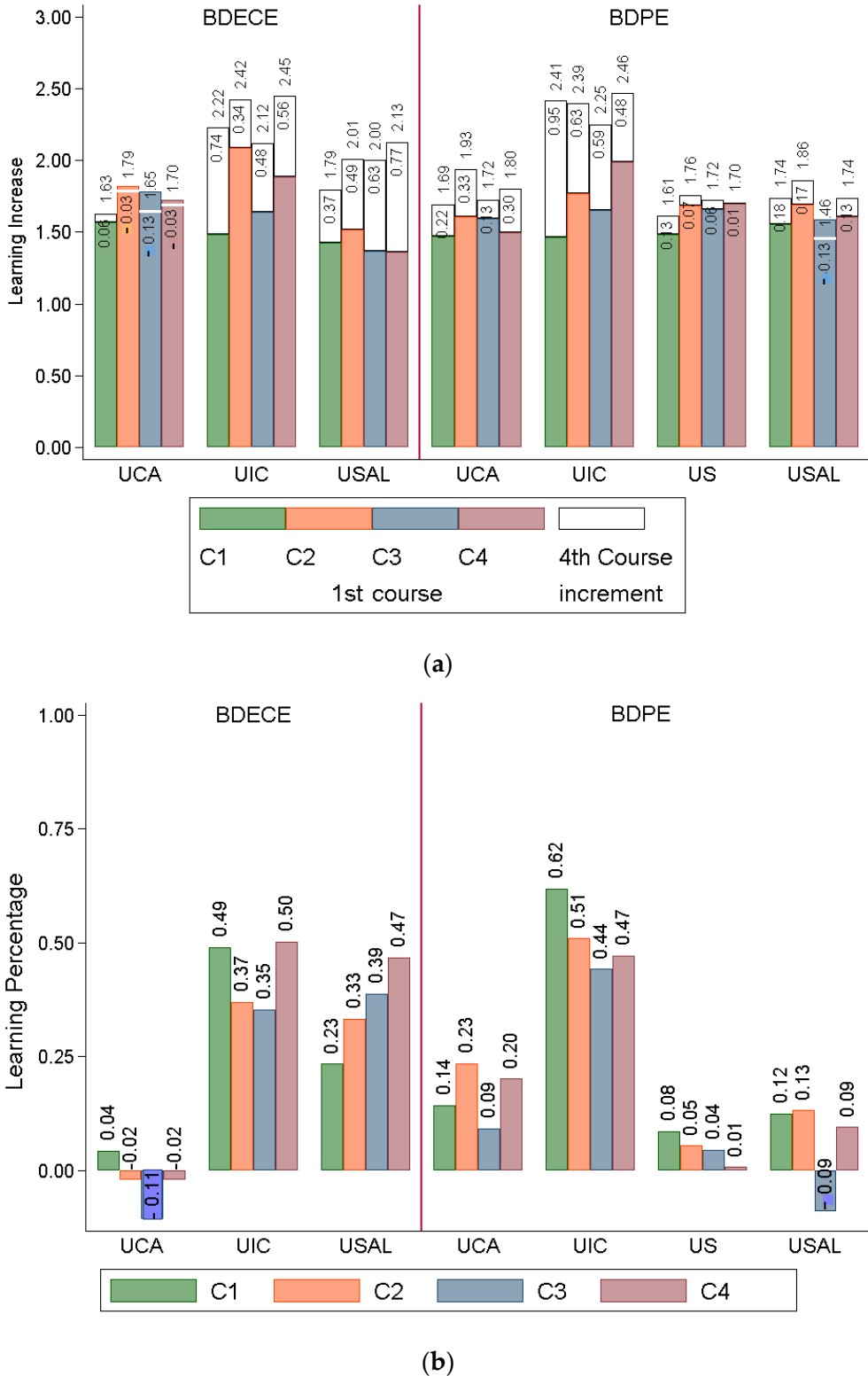

(**a**)

(**b**)

**Figure 7.** Learning Increase (**a**) and Learning Percentage (**b**) declared by the students in each competency in the BDECE and BDPE degrees according to university.

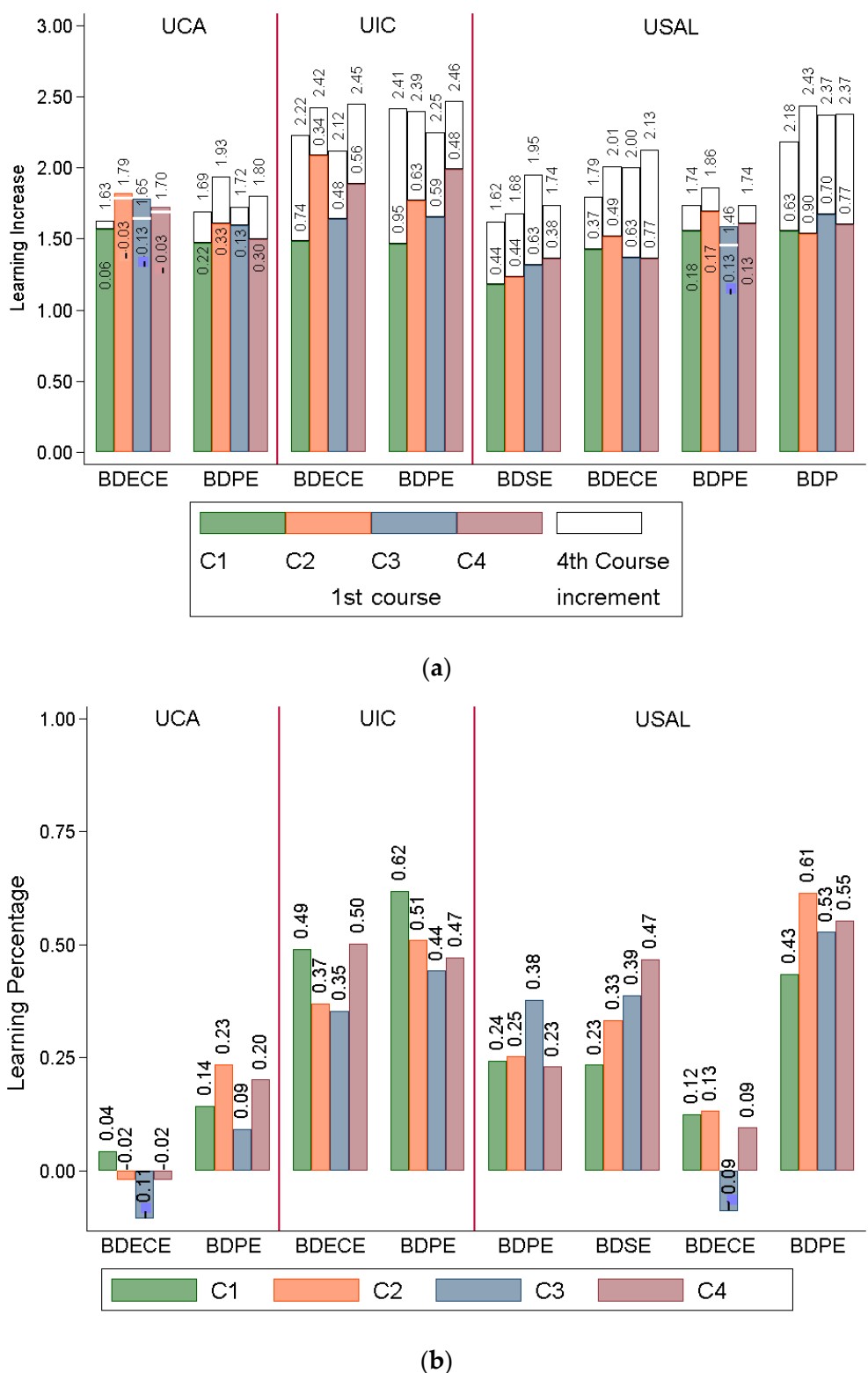

**Figure 8.** Learning Increase (**a**) and Learning Percentage (**b**) declared by students in the four competencies in each university according to the degree.

In order to test whether or not the differences on Learning Increase between first and fourth year students are statistically significant, two factorial ANOVAs are developed. While the first statistical model disposes Competencies' Learning Increase as dependent variable, the second model disposes Competency Units' Learning Increase. In both course, university, degree and interaction terms are disposed as independent variable, as well as

competency/competency units. Factorial ANOVAs are chosen as they enable measuring differences on one variable among several independent variables. The first model is shown in Table 4 and suggests that, although differences in Competencies' Learning Increase among several independent variables are statistically significant, the most relevant factor for changes in Learning Increase is course, as it displays the strongest significant effect size (F = 597.87; $p$ = 0.000; $\eta^2$ = 0.568). Course-differences across Universities and Degrees are also statistically significant.

**Table 4.** First factorial ANOVAs Model.

| Variables | df | MS | F | $p$ | Effect Size $\eta^2$ |
|---|---|---|---|---|---|
| Model | 31 | 39.474 | 57.19 | 0.000 | 0.796 |
| Course | 1 | 13.313 | 597.87 | 0.000 | 0.568 |
| Competency | 3 | 1.093 | 16.36 | 0.000 | 0.098 |
| University | 3 | 7.891 | 118.12 | 0.000 | 0.438 |
| Degree | 3 | 4.644 | 69.52 | 0.000 | 0.315 |
| Course x Competency | 3 | 0.066 | 0.98 | 0.399 | 0.006 |
| Course x University | 9 | 0.643 | 3.21 | 0.001 | 0.060 |
| Course x Degree | 9 | 0.575 | 2.87 | 0.003 | 0.054 |
| Residual | 454 | 10.109 | | | |

N = 486; df = Degrees of freedom; MS = Mean squares.

The second model's results converge with the previous one. As is shown in Table 5, differences in Competency Units are also significant among many independent variables. However, Competency Units is the factor displaying the strongest significant effect size over Learning Increase (F = 597.87; $p$ = 0.000; $\eta^2$ = 0.570). Both models point out the strength of the findings previously analyzed.

**Table 5.** Second factorial ANOVAs Model.

| Variables | df | MS | F | $p$ | Effect Size $\eta^2$ |
|---|---|---|---|---|---|
| **Model** | 47 | 0.959 | 35.82 | 0.000 | 0.796 |
| **Course** | 1 | 15.527 | 579.70 | 0.000 | 0.570 |
| **Competency Units** | 5 | 0.513 | 19.14 | 0.000 | 0.179 |
| **University** | 3 | 3.227 | 120.49 | 0.000 | 0.452 |
| **Degree** | 3 | 1.719 | 64.18 | 0.000 | 0.305 |
| **Course $\times$ Competency** | 5 | 0.033 | 1.23 | 0.292 | 0.014 |
| **Course $\times$ University** | 15 | 0.162 | 6.06 | 0.000 | 0.172 |
| **Course $\times$ Degree** | 15 | 0.056 | 2.10 | 0.009 | 0.067 |
| **Residual** | 438 | 0.027 | | | |

N = 486; df = Degrees of freedom; MS = Mean squares.

## 4. Discussion

*4.1. Question Q1: How Much Do Students from Spanish University Education Degrees in Spain Improve their Sustainability Competencies During Their Studies?*

Figure 1 shows that there are no great variations in the learning declared in the four competencies, either Learning Increase or Learning Percentage. First-year students declare that they know approximately 50% of what they should know at the end of their studies in the four competencies (1.6 out of 3 on average), while fourth-year students declare that they have learned 66% (1.98 out of 3) (Figure 1a).

On average, the Learning Percentage of fourth-year students is 27% (Figure 1b). Learning in all competencies is similar to the average, except for C3-participation in community processes, where students report the least Learning Percentage (23%). Training in teaching techniques that promote participation and collaboration strengthens ethical values such as responsibility, solidarity and commitment to sustainability, in addition to

promoting the development of critical thinking. Better training in teaching techniques for active, participatory, and collaborative learning would report higher levels of student engagement and achievement. As stated by Filho et al. [30], academics should develop collaborative approaches and appreciate the multicultural vision of sustainability, especially in education degrees, as their graduates are the future teachers of new generations of citizens and can act as catalysts for socio-environmental change and transformation [20], thus contributing to the achievement of fairer, more sustainable and balanced societies.

As shown in Figure 2, the learning declared by CU, both Learning Increase and Learning Percentage, is similar in the two CUs of C1-Critical contextualization of knowledge. In C4-Ethics, on the other hand, CU4.2-Promotes education in values yields better results than CU4.1-Is consistent in actions. In fact, CU4.2 obtains the best learning values of the six CUs analyzed, both in Learning Increase (2.17 out of 3) and Learning Percentage (33%).

It seems that CU4.2 is not only the CU in which students feel better prepared upon entering college, but it is also the CU on which education teachers focus much of their efforts. In teacher-training carried out at these education levels, citizenship education is considered a key aspect [31]. This seems to be clearly reflected in the survey results.

Therefore, in response to the first research question, it seems that students improve all their sustainability competencies throughout their studies. However, they state that they have achieved less than 30% of the learning that they should achieve in practically all CUs (with the exception of CU4.2). We consider that this value is very low and shows that the sustainability competencies are not being adequately developed in the education degrees analyzed.

### 4.2. Question Q2: Is the Improvement Homogeneous at All Domain Levels?

As shown in Figure 2, both Learning Increase and Learning Percentage present similar levels at all domain levels for all competencies, except C3-Participation in community processes. The domain level L1 of C3 is the level in which the students declare they have learned least, both in Learning Increase and Learning Percentage. The low Learning Percentage obtained at the L1 level (13%) explains that C3 is the competency with the lowest Learning Percentage in general (23%), as shown in Figure 1b.

Likewise, the low level of learning declared by first-year students at level L1 of C3 (1.42 out of 3) indicates that they do not know of any community educational programs that promote participation and commitment in socioenvironmental improvement. This result is contradictory, because the students also declare that they know how to function satisfactorily in community educational projects, promoting participation (level L2 of C3, 1.8 out of 3) even above that of the Learning Increase declared in the levels of the other competencies.

The low learning achieved at level L1 of C3 can be understood as a failure in the development of this competency in the Education degrees, since one of the specific competencies to be developed in these Degrees is precisely "knowing ways of collaboration with the different sectors of the educational community and the social environment". It makes sense that first-year students do not feel competent in C3. However, the fourth-year students should show learning in this competency since it is evaluated in the "Curricular Practices" subjects. These subjects are evaluated by external tutoring teachers at the school or university where the students study, using standard indicators such as:

- The student knows the social and educational institutions with which the school or university interacts.
- The student knows of the collaboration conducted by the university with the community and the environment, and participates in them.
- In their teaching and tutorial performance, the students use the resources deriving from institutional collaboration.

This result is consistent with the conclusions of other studies [32,33] that show that there is little or no transfer from theory to practice in teacher training. The results obtained show, for competency C3, the gap that exists between theory and practice. According

to these results, students perform in the C3 competency without having a solid basic training. Korthagen [34] affirms that this is feasible, and that people do not act solely on the basis of a logical and rational analysis, but that the teaching profession is especially influenced by fear, lack of certainty and lack of stability. These feelings eliminate any rational intention [34], and can lead to this type of mismatch between domain levels in an eminently practical competency that involves other groups outside the student with whom the student interacts.

In response to the research question, the improvement identified is homogeneous for all domain levels, except in the case of C3, where the L1 level is less developed than the others.

*4.3. Question Q3: Is the Improvement Homogeneous in All the Degrees Analyzed?*

Figure 4 shows that the BDP is the degree in which students declare the best results by far in all competencies. The declared Learning Percentage stands out in C2-Sustainable use of resources (61%). C1-critical contextualization of knowledge is the competency in which students declare the worst Learning Percentage outcomes (43%), although this learning is superior to that declared in any of the competencies of any other degree. This may be due to the fact that we only have data from the BDP of the USAL, and only 14.3% of students in the fourth year responded to the survey, while 78.6% of the first-year respondents answered the questionnaire (see Table 2). Given such a low sample in the fourth year, the results may not be representative, or it may be that the more motivated fourth-year students responded [25].

On the other hand, when comparing the Learning Percentage in Figure 4b, it is observed that the greatest learning is achieved in different competencies, depending on the degree analyzed. Thus, the BDSE achieves the best learning outcomes in C3-Participation in community processes, the BDECE in C4-Ethics, the BDPE in C1-critical contextualization of knowledge (with similar values in C2) and the BDP in C2-Sustainable use of resources. These results coincide with those presented in other studies [35,36], which indicate that the learning and development of cognitive skills in students varies depending on their academic specialty due to the processes of interaction with teachers.

In Figure 5 we will analyze the C1 and C4 competencies since they are the only competencies with two CUs. The analysis of C2 and C3 has already been conducted in Figure 4.

In BDSE and BDP, better Learning Percentage is clearly obtained in CU1.2-Possesses critical thinking and creativity than in CU1.1-Understands the functioning of natural, social, and economic systems. However, the two CUs of C1 obtain similar results in the BDECE and the BDPE. These results are logical because in the BDSE and BDP the aim is to train the student (1) in a professional profile with vocational orientation towards educational issues; (2) with a predisposition for analysis, reflection and creativity; and (3) with a critical spirit and concern for transformation and social changes [37].

Learning C4-Ethics has a long tradition in education degrees. Students report greater Learning Increase in CU4.2-Promotes education in values than in CU4.1-Is consistent in actions in all degrees, and greater Learning Percentage in three: BDSE, BDPE and BDP. In BDECE, future preschool teachers report a similar Learning Percentage in the two CUs of C4 (approx. 32%). This can be explained by the awareness of this group regarding the coherence of their personal actions with respect to and appreciation of diversity (CU4.1). This awareness has a decisive influence on the development of the learners to whom their professional activity will be directed (in this case, children in early childhood education). These results are consistent with other research carried out on the curricula of education degrees [20].

The analysis of Figures 4 and 5 leads us to conclude that the improvement identified is not homogeneous in all the degrees analyzed. Each degree seems to focus on further developing some specific competencies and competency units. Table 6 summarizes these results.

**Table 6.** Competency units with greater learning declared by students at each degree.

| Degrees/CU | CU1.1 | CU 1.2 | CU 2.1 | CU 3.1 | CU 4.1 | CU 4.2 |
|:---:|:---:|:---:|:---:|:---:|:---:|:---:|
| BDSE | | | | X | | X |
| BDECE | | | | | X | X |
| BDPE | X | X | X | | | X |
| BDPE | | | X | | | X |

As may be seen in Table 6, all the degrees stand out in the development of the CU4.2-Promotes education in values, and focus on some other competency unit. We may therefore conclude that "Promoting education in values oriented to the formation of responsible, active and democratic citizens" is the common feature of the development of sustainability in the four degrees.

*4.4. Question Q4: Is the Improvement Homogeneous in All the Universities Analyzed?*

Figure 6 shows that the US and the UCA are the universities in which the students declare the worst Learning Percentage. The UCA obtains maximum learning (just 12%) in C2-Sustainable use of resources, and the US a maximum of only 8% in C1-Critical contextualization of knowledge. At the UCA, students declare unlearning (negative learning) in C3-Participation in community processes (−0.001). In other words, fourth-year students report less learning than first-year students. This is probably due to the fact that first-year students are affected by the Kruger-Dunning effect [38], a cognitive bias whereby individuals with low skill or knowledge experience an illusory feeling of superiority and incorrectly measure their ability (above the real valuation).

The UIC is the university in which the students declare that they achieve the highest learning, both Learning Increase (equal to or greater than 2.18 out of 3) and Learning Percentage (varying between 40% for C3-Participation in community processes and 55% for C1-Critical contextualization of knowledge).

The USAL achieves similar Learning Percentage in all competencies, ranging from a minimum of 26% in C1-Critical contextualization of knowledge to a maximum of 34% in C4-Ethics.

As is clear from the previous paragraphs, the learning declared by the students is not homogeneous in the universities analyzed, since significant differences exist between them. It should be noted that the improvement observed in all the degrees in CU4.2-Promoting education in values is not observed when the data is analyzed at the university level.

Morland-Painter et al. highlight that 'integrating sustainability into the curriculum must be closely aligned with systemic institutional integration' [39]. However, in universities, the willingness of policy-makers and decision-makers to move towards a sustainable future is usually insufficient [40]. According to Lozano et al. [41], 'in spite of a number of sustainable development initiatives and an increasing number of universities becoming engaged with sustainable development, most higher education institutions continue to be traditional, and rely upon Newtonian and Cartesian reductionist and mechanistic paradigms'.

*4.5. Question Q5: Does the Same Degree Provide Homogeneous Learning in All the Universities Where It Is Taught?*

Figure 7 shows that the BDECE clearly obtains worse results in the UCA than in the UIC and the USAL. In fact, the Learning Percentage in three of the four competencies is negative (C2, C3 and C4). Between UIC and USAL, the main difference is in the Learning Percentage declared in C1, which has worse results in USAL (23% vs. 43% for the UIC). In general terms, UEC BDECE students are those who declare that they achieve more learning.

In the BDPE, the UIC clearly achieves better results than the other universities for all competencies, both in Learning Increase and Learning Percentage. UCA, US, and USAL students report very similar Learning Increase across the four competencies, but there are

significant differences in Learning Percentage. The US is the university with the worst results (between 1% and 8%), followed by the USAL, which achieves between 9% and 13% of Learning Percentage (with the exception of C3-Participation in community processes, in which a negative learning of 9% is declared by students). Finally, UCA students report more learning in C2-Sustainable use of resources and C4-Ethics (23% and 20% respectively), and worse results in C1-Critical contextualization of knowledge and C3-Participation in community processes (14% and 9% respectively).

The Kruger-Dunning effect could probably explain the negative learning of 2% declared by students in C2 and C4 in the BDECE of the UCA, but it can hardly justify the −11% declared in C3, or the −9% declared also in C3 the BDPE of the USAL. Both percentages are very high in absolute value, and both occur in C3. The number of students in the sample, 51 in the first year (81%) and 26 in the fourth year (22%), does not seem to be the reason for this negative Learning Percentage at USAL, since the other three competencies show positive Learning Percentage. between 9% and 13%. In the BDECE of the UCA, 37 first-year students (20.7%) and 61 fourth-year students (70.3%) responded to the questionnaire and, in the Learning Percentage declared in the four competencies C3 stands out clearly. Thus, it seems that there is indeed a problem with competency C3, and if the Kruger-Dunning effect occurs in first-year students, then C3-Participation in community processes is not really being developed as it should.

In the case of the UIC, which presents the best results in both degrees, the questionnaire was answered in the BDECE and the BDPE by 17 and 18 first-year students (51.5% and 40.9%, respectively), and by 14 fourth-year students in both degrees (37.8% and 27.5%, respectively). The low number of students, the smallest in the entire sample, might account for the good results if, for example, they were especially motivated students. However, the response percentages are similar or even higher than those obtained in other degrees. This effect is also observed in the results of the BDECE of the USAL, in which only nine fourth-year students (10.1%) answered the questionnaire and good results were also obtained in the four competencies. In this case, it does seem more likely that the sample size has a certain influence on the result.

The analysis of the results confirms that the learning that students claim to achieve in each competency is influenced by the disciplinary content that students develop throughout their studies [36].

In answer to the research question, the same degree presents different learning in all the universities where it is taught.

*4.6. Question Q6: Does the Same University Provide Homogeneous Learning in All the Degrees It Teaches?*

As shown in Figure 8, the two degrees analyzed at the UCA, BDECE and BDPE, obtain similar results for Learning Increase, but are very different in Learning Percentage. Learning Percentage is also very low, especially in BDECE, in which three competencies present negative learning. Clearly, there is no institutional strategy at UCA to develop sustainability.

At the UIC, BDECE and BDPE achieve similar learning, both Learning Increase and Learning Percentage, which is compatible with the UIC having an institutional policy to develop sustainability.

USAL does not seem to have a defined sustainability strategy either, since there are notable differences in Learning Percentage among its four degrees. The degree with the worst results is BDECE. These results are consistent with those presented in [25].

In response to the research question, the answer is: "it depends on the university".

Taking into account everything above, it seems that the university where the degree is taught has more influence on the results in terms of sustainability than the degree, except perhaps for C4-Ethics, which is the competency that is clearly developed more in all degrees, regardless of the university in which it is taught.

These results are probably due to the coexistence of different visions concerning the study of the quality of education. In recent years, the search for efficiency in education

has fostered a managerial vision of the university. From this perspective, universities are considered professional knowledge organizations where academics are judged for the quality of their work. However, the resistance to change offered by some higher education institutions is also known, especially those that have been in existence for several centuries [20] and are aligned with a more "philosophical and humanistic" vision of education.

### 4.7. General Discussion

The results presented in this paper show that ESD is not being achieved in Education degrees at Spanish universities. Graduates are not being adequately trained in the four sustainability competencies that the CRUE indicates should be developed in all degrees of the Spanish University System. On average, graduates achieve approximately two-thirds of the learning they should achieve at the end of their studies. From the point of view of the degrees themselves, students learn only 27% of what they should have learned. None of the four competencies stands out positively or negatively, so the learning problem in ESD is generalized, and not limited to certain competencies or domain levels.

However, not all degrees have the same behavior, and some degrees perform better than others. In particular, the bachelor's degree in Pedagogy achieves much better results than the other analyzed degrees. Some of the future work to be carried out by the authors consists of analyzing what activities are undertaken by bachelor's degree in Pedagogy students in order to achieve such outstanding results in ESD when compared with the other Education Degrees. At university level, the learning achieved in ESD by UIC students stands out positively over that achieved by students from other universities. As future work, the authors intend to study what characteristics the degrees of the UIC possesses that are lacking in the degrees of other universities. It may be that the idiosyncrasies of the UIC itself, or the fact that it is a private university and that the rest of the universities analyzed are public universities, are determining factors.

University education plays a key role in the development of more sustainable societies [42]. In this context Education degrees have special relevance, since the training of teachers, future education professionals, is a powerful tool for change and social transformation. According to Korthagen [34], the ideal process to promote critical and reflective learning (which bridges the gap between "theory" and "practice") should be based on an alternation between action and reflection. In this sense, the promotion of a dialogic education, linked to real-world experiences, in Education degrees is regarded as an optimal approach for teaching and learning sustainability [13]. The inclusion of active methodologies in the subjects, in direct connection with the problems of the real world, favors the generation of shared spaces for reflection and democratic participation, as well as creating and disseminating new knowledge [43]. Part of the future work to be carried out by the authors will consist of analyzing whether bachelor's degree in Pedagogy students or UIC students learn by using this model.

### 4.8. Limitations of This Work

The sample size is small for the nine degrees. In addition, the sample of students is very uneven between the degrees and, even within a degree, there is a great disparity between the number of first- and fourth-year students. The degrees with fewer answers seem to achieve better results than the others, which would suggest that the number of answers is not significant enough to draw conclusions. The analysis in Table 2 could reveal the existence of a bias in the sample, whereby the most motivated and the highest learning students are those who tend to answer the questionnaire.

Furthermore, the questionnaire measures students' perception of their sustainability competencies at a specific time, not their actual competencies. As a consequence of all the foregoing, the results of this study cannot be extrapolated to the rest of education degrees in the Spanish university system, although the methodology used in this study can be used

by other researchers to replicate the work in their respective universities and to compare their results with those presented in this work.

## 5. Conclusions

In this paper, an analysis is conducted of the perception that the students of nine Spanish university education degrees have about their training in sustainable development. The starting hypothesis is that fourth-year students have improved their sustainability competencies compared to first-year students. The methodology is applied to a case study: nine Higher Education Degrees.

The analysis is performed on the basis of the students' responses to a questionnaire. To analyze the responses, two composite indicators have been created: Learning Increase, which measures the learning perception of each student with respect to the item analyzed, and Learning Percentage, which measures the relationship between the learning perceived in each item by the students and the learning that could have been achieved on completion of their studies.

Six research questions have been formulated and analyzed using the two composite indicators. Regarding the first question—How much do students from Spanish university education degrees in Spain improve their sustainability competencies during their studies?—it is found that students improve all their sustainability competencies throughout their studies. However, they state that they achieve only 27% of the learning they should achieve. The answer to the second question—Is the improvement homogeneous at all domain levels?—is that the improvement identified is homogeneous for all the domain levels of the taxonomy except for C3-Participation in community processes, in which a lower achievement of learning is observed at the L1 level of the taxonomy ("Know") than at the other two levels of the taxonomy ("Know how" and "Demonstrate and Do"). Questions 3 and 4 are about whether the improvement is homogeneous in all the degrees and universities. The results show that the improvement depends on both the degree and the university, although in all the degrees the students state that they perceive an improvement in C4-Ethics and, in particular, in "Promoting education in values oriented to the formation of responsible, active and democratic citizens", although this improvement is not perceived in the analysis carried out at the university level. It is interesting to observe that the students of each degree declare that they have achieved a greater learning in a different competency. In the BDSE, this is C3-Participatory processes; in the BDECE it is C4-Ethics; in the BDPE it is C1-Critical contextualization of knowledge (with similar values in C2), and in the BDP it is C2-Sustainable use of resources. The answer to question 5 is that the same degree presents different learning in all the universities where it is taught, and the answer to question 6—Does the same university present homogeneous learning in all the degrees it teaches?—is "it depends on the university".

The results presented in this work are consistent with those in other published works, both in regard to the students' opinion on the sustainability learning they receive at university and with the content of the degree curricula.

According to the opinion of the students about their sustainability training, all the university education degrees analyzed in this work develop professional ethics in part, although the results they achieve are far from those expected (students only learn 27% of what they should learn). The learning that they declare to have achieved in the rest of the competencies analyzed is very different depending on the degree and the university, but in any case it is very far from what they should achieve. The competency in which students achieve the worst results is C3-Participation in community processes, in which they learn only 23% of what they should learn. In view of these results, it is essential to improve the Education for Sustainable Development received by students on university degrees in Education in Spain, since they will be the teachers who will train the professionals of the future.

**Author Contributions:** F.S.-C.: Conceptualization; Data curation; Funding acquisition; Investigation; Methodology; Project administration; Resources; Supervision; Validation; Visualization; Roles/Writing—original draft; review &editing. F.M.M.-P.: Conceptualization; Investigation; Methodology; Validation; Visualization; Roles/Writing—original draft. D.R.-P.: Conceptualization; Data curation; Formal analysis; Methodology; Software; Visualization; Roles/Writing—original draft. B.S.: Conceptualization; Investigation; Methodology; Validation; Visualization; Roles/Writing—original draft. All authors have read and agreed to the published version of the manuscript.

**Funding:** This work was supported by the Spanish Ministerio de Economía y Competitividad, from study design to submission, under grant number EDU2015-65574-R; by the Spanish Ministerio de Ciencia, Innovación y Universidades, the Spanish Agencia Estatal de Investigación (AEI), and the Fondo Europeo de Desarrollo Regional (FEDER), from study design to submission, under grant number RTI2018-094982-B-I00.

**Institutional Review Board Statement:** All the results presented in this work satisfy the "Ley Orgánica 3/2018, de 5 de diciembre, de Protección de Datos Personales y garantía de los derechos digitales" (LOPD) of Spain.

**Informed Consent Statement:** Informed consent was obtained from all subjects involved in the study.

**Data Availability Statement:** Not applicable.

**Acknowledgments:** We want to thank all the researchers who have participated in the EDINSOST project for their work. We also want to thank the students who have answered the questionnaire.

**Conflicts of Interest:** The authors declare that they have no conflict of interest.

## Appendix A

Below is the questionnaire answered by the students, as published in [14]. The answers are classified according to a Likert scale of 4 points with the following meaning: Strongly disagree, Disagree, Agree and Strongly agree.

1. I know the interrelation between natural, social and economic systems.
2. I analyze and understand the relationships between natural systems and social and economic systems.
3. I anticipate the repercussions of changes in natural, social and economic systems.
4. I know procedures and resources to integrate sustainability in the subjects.
5. I analyze the opportunities presented in the subjects to plan educational projects to integrate sustainability.
6. I design educational projects from the perspective of sustainability.
7. I identify the possible socio-environmental impacts derived from my educational activities.
8. I know how to develop educational actions that minimize negative socio-environmental impacts.
9. I design and develop educational actions in which I take into account the negative socio-environmental impacts and I incorporate corrective actions.
10. I know community educational programs that encourage participation and commitment in socio-environmental improvement.
11. I know how to develop myself satisfactorily in community educational projects, encouraging participation.
12. I design and carry out socio-educational activities in participatory community processes that promote sustainability, feeling myself an integral part of my environment.
13. I know the ethical principles of sustainability.
14. I understand and integrate the ethical principles of sustainability in my professional and personal actions.
15. I design and/or manage educational projects taking into account ecological ethics, to improve the quality of life and promote the common good.
16. I consider the promotion of sustainable human development as a fundamental purpose of citizen education.

17. I critically analyze and value the consequences that my personal and professional performance may have on the integral development of students and on the promotion of sustainable human development.
18. I design and develop educational intervention proposals that integrate sustainability values and result in justice and the common good.

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
