# Peer review of "Education for Sustainable Development in Spanish University Education Degrees"

_sustainability, doi:10.3390/su13031467_

Round 1
Reviewer 1 Report
Comments:
- Lines 34-36, 39-42 must bring appointments.
- The objective of this work is not adequately defined.
- What are the starting hypotheses?
- Why are these 6 questions raised and not others? What literature are they based on? Are there studies that have answered these in your context?
- It would be interesting to list other limitations that have been found, in addition to what is indicated in 3.7, for example: what limitations does Google Forms have? Is the questionnaire used a reliable instrument?
Reviewer 2 Report
Dear authors:
Research is properly, but discussion could be improved including more researches. Moreover, in the following sentence "These results are consistent with those presented in other works [20]." you wrote "other works", but you only present one of them.
Best regards.
E.
Reviewer 3 Report
This paper addresses a key issue in the study of sustainability: the analysis of the training of future educators in this field.
Given that education has the potential to generate change in future generations, teacher training is of great importance.
The paper is well focused, but in my opinion, it needs some changes that I expose next:
- Introduction.
1.1. The introduction is too short. It does not provide enough theoretical background to frame the content of the paper.
It needs, at least, a more in-depth review of previous studies that have analyzed the training of future educators in Spain or internationally.
See for example the research synthesis:
Stevenson, R. B., Lasen, M., Ferreira, J. A., & Davis, J. (2017). Approaches to embedding sustainability in teacher education: A synthesis of the literature. Teaching and Teacher Education, 63, 405-417. doi: 10.1016/j.tate.2017.01.013
Or the studies from Spanish context:
Sureda-Negre, J., Oliver-Trobat, M., Catalan-Fernández, A., & Comas-Forgas, R. (2014). Environmental education for sustainability in the curriculum of primary teacher training in Spain. International Research in Geographical and Environmental Education, 23, 281-293. doi: 10.1080/10382046.2014.946322
Varela-Losada, M., Arias-Correa, A., Pérez-Rodríguez, U., & Vega-Marcote, P. (2019). How Can Teachers Be Encouraged to Commit to Sustainability? Evaluation of a Teacher-Training Experience in Spain. Sustainability, 11, 4309. doi: 10.3390/su11164309
1.2. It is also necessary to explain in more detail the importance of analyzing teacher training.
- Instrument.
2.1. As the description of the questionnaire is currently drafted, its content is not clear.
What is the exact content of the items? With respect to what should students indicate their degree of agreement? Is it about expressing your degree of agreement with the degree of training that they have acquired in their studies?
According to what is indicated in the abstract, it seems that it is so. But the description of the instrument is not sufficiently specified.
2.2. The domain levels must be also explained in this section.
- Methodology.
Lines 157-159. It's suggested to specify the exact values for KMO, Bartlett sphericity test and Cronbach’s alpha coefficients.
With respect to Cronbach’s alpha coefficients, 0.6 is not a high value. If the alpha values are near to 0.6, it's suggested to indicate it in the limitations of the study.
- Results.
4.1. I think the focus of the statistical analysis of the results should be changed.
As the comparison between 1st and 4th year students is being made right now, it can simply be appreciated globally if there is a perception of increased knowledge.
But it cannot be known whether this increase reaches a level of statistical significance or not. In other words, it cannot be said whether the differences between 1st and 4th year students are statistically significant or not.
Statements such as "there are no great variations in the learning declared in the four competencies" do not provide exact information. They only report on the trend of the results.
Have the authors thought about developing MANOVAs that allow the means between 1st and 4th year students to be compared; between the scores of the different competences; or between universities?
A MANOVA would allow knowing if the differences in the scores are statistically significant or not; and in this way the information provided in the results would be richer.
4.2. I think that the section 3. Results and discussion should be divided in two different sections, one for results, and the other one for discussion.
I mean, the results section should just contain the results itself, and its explanation. But not its interpretation or discussion (this content must be relocated to discussion section).
For example, some lines that should be relocated are: 223-230; 246-254; 277-290; 308-311 (and maybe the lines explaining the Kruger-Dunning effect are the best example of content that should be relocated).
4.3. Table 5 seems constructed based on qualitative conclusions.
It's suggested to construct a table with numerical data that support these conclusions.
- Discussion.
The discussion should include a brief explanation about the practical implications of results. I mean, it should be briefly explained some general lines on how universities could improve the sustainability training of future education professionals.
- Conclusions.
The information about the sample (line 473) should be removed from this section. This is not a conclusion.
Also, the information about the instrument could be reduced, offering just the essential information to understand the rest of the section.
- Formal issues.
7.1. There are some misprints. The text should be reviewed to detect and correct them.
For example, in the abstract (line 24), "declare an(d) improvement".
7.2. Tables 2 and 3 can be merged. This saves space and provides greater clarity for the reader.
7.3. The figures allow you to visually appreciate the trends of the results. However, such a large number of figures can be excessive.
Have the authors thought of unifying figures 1a and 1b? and, 2a and 2b...?
It could only be done by adding a value to figures a (for example in parentheses).
Actually, the visual information of figures "b" is already included in figures "a", since the white region of each bar of figures "a" is proportional to the full bar of figures "b".
Round 2
Reviewer 1 Report
Dear authors,
The revised manuscript has improved after the last corrections, although I continue to observe some weaknesses. I comment on them below:
- Discussions that confront the results obtained with the experts in the field should be included. In subsections 4.2, 4.3, 4.4 and 4.5 there are no bibliographic references that provide arguments and replicate the results obtained.
-It is recommended to review the keywords to obtain a better impact of the paper.
Are the conclusions consistent with the evidence and arguments presented?
Reviewer 3 Report
I congratulate the authors for the improvements made to the article.
Before ending the job, I suggest just two more changes:
1. KMO and sphericity test.
These statistics do not need to be provided for all scales. Usually only the value of these statistics is indicated for the whole questionnaire, as a requirement to perform the factor analysis.
2. Reporting data.
When probability "p" values and statistics ranging from 0 to 1 are reported (such as Cronbach's alpha coefficient), only decimals are indicated. That is, it is not "p <0.01", but "p <. 01".
